# Video decoding from human fMRI data with a multi-stream sensory approach

**Matteo Ferrante**
Department of Biomedicine and Prevention
University of Rome, Tor Vergata
`matteo.ferrante@uniroma2.it`

**Matteo Ciferri**
Department of Biomedicine and Prevention
University of Rome, Tor Vergata

**Nicola Toschi**
Department of Biomedicine and Prevention
University of Rome, Tor Vergata
Martinos Center For Biomedical Imaging
MGH and Harvard Medical School (USA)

**Editors:** Marco Fumero, Clementine Domine, Zorah Lähner, Donato Crisostomi, Luca Moschella, Kimberly Stachenfeld

## Abstract

In this study, we present a novel multi-stream sensory approach for decoding video stimuli from human fMRI data. Leveraging a dataset of 1,000 short video clips and associated fMRI data, we explore the integration of visual, textual, and audio modalities to enhance the accuracy of brain decoding models. We develop subject-specific encoding models that predict brain activity from modality-specific embeddings and apply functional alignment across subjects to improve model generalization. Our decoding framework employs Ridge regression within identified regions of interest for each modality, followed by a retrieval process based on Euclidean search. The results demonstrate that integrating multiple sensory streams significantly enhances the performance of decoding models, with the combined Video+Text+Audio modality achieving the highest identification and retrieval accuracy.

## 1 Introduction

Vision is one of the primary modalities through which we interpret the external world, involving complex dynamics related to movement, object recognition, tracking, and multisensory integration. Understanding how the brain processes this information is a heavily researched yet still not fully understood area. Recent advancements in brain encoding and decoding using non-invasive techniques like EEG, MEG, and fMRI have led to the development of computational models that map external stimuli to brain representations and vice versa, and the availability of large public fMRI datasets and multimodal foundation models has facilitated these advancements [18]. Typically, encoding involves using a pretrained model to generate modality-specific embeddings , which are then projected onto brain activity via linear mapping techniques. For decoding, linear or non-linear models project brain activity measured in concomitance with the stimuli into an embedding space representing the latter. These embeddings can be used for tasks like retrieval—identifying the stimulus linked to a specific brain activity pattern —or reconstruction of the stimuli themselves using generative models. Recent literature has shown significant progress in fMRI-based image decoding [21, 3, 10, 12, 23, 19],

Proceedings of the II edition of the Workshop on Unifying Representations in Neural Models (UniReps 2024).

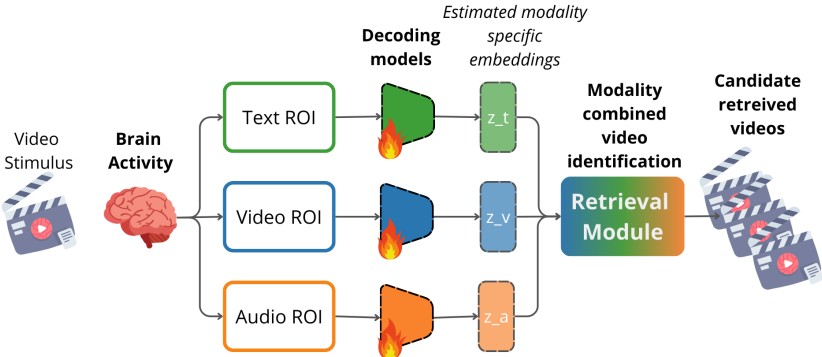

**Decoding Pipeline**

Figure 1: Decoding pipeline that translates brain activity into modality-specific embeddings to retrieve the corresponding video stimuli. Brain activity data is processed through modality-specific ROIs for text, video, and audio. Decoding models, trained with Ridge regression, estimate embeddings for each modality ($z_t$, $z_v$, $z_a$). These embeddings are then combined in a retrieval module to identify and output the candidate videos that best match the estimated embeddings, effectively decoding the original video stimulus.

particularly for image retrieval and reconstruction. Most methods are variations of the following concept: the brain creates a representation of external stimuli, while we can obtain external stimulus representation (i.e. image embeddings) using a computational model, then learning a mapping between these two representations completes the decoding pipeline. These methods differ e.g. in the generative models used, in how the models are conditioned, and in the techniques used to compute the mapping between brain activity and embeddings—ranging from linear layers to neural networks. However, they all revolve around the central idea that the brain computes something analogous to image embeddings (ideally a representation in a manifold homomorphic the the model's one) which can be captured via fMRI measurements, and that with sufficient data, a mapping between brain activity and computational model-derived embeddings can effectively link brain activity to external stimuli. This concept also underpins language encoding and decoding [1, 24, 15], where large language model embeddings serve as surrogates to approximate, through linear layers, the language processing which occurs in the brain during listening and comprehension. Similar approaches have shown promising results in decoding music from brain activity [8, 13]. The closest related work in this domain is represented by [4, 22], which directly addresses the problem of video reconstruction from fMRI data using a different dataset from the on employed here [26]. This latter dataset comprises 18 training videos (each 8 minutes long) and 5 test videos of the same length, collected from 3 subjects. The approach in [4, 22] is based on subject-specific semantic mapping between fMRI data and Contrastive Language Image Pretraining (CLIP) embeddings, along with attention based modules to condition generative models for generating temporally coherent images to reconstruct videos. In this work, we approach video decoding from a different perspective. We use a rich dataset [6] of 1000 short video stimuli and concomitant fMRI data to build cross-subject models [11] for decoding through video retrieval from fMRI data. We hypothesize that video processing in the brain can be decomposed into three distinct streams: a visual stream (recognizing shapes, patterns, and objects in images), a semantic stream (understanding what is happening in the video), and an audio stream, which provides multisensory integration that aids in video comprehension. Based on this hypothesis, our pipeline consists of three main components: First, we construct subject-specific encoding models that predict brain activity from modality-specific embeddings (video, text, and audio) to identify responsive brain regions (regions of interest, ROIs) for each modality. Next, we perform functional alignment [14] to create a robust training and testing set across subjects. Finally, we develop a set of modality-specific decoding models that estimate embeddings from brain activity, which can be used for video retrieval. We demonstrate how multistream integration enhances decoding performance and provide examples at `https://mind2music.my.canva.site/decoding-video-nips-sito`. Figures 1 and 2 depict our decoding and encoding frameworks, respectively.

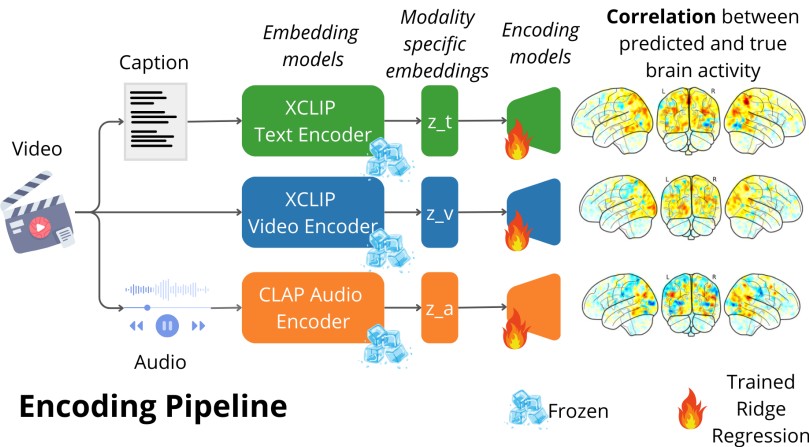

Figure 2: This figure shows the encoding pipeline that processes video data into modality-specific embeddings and maps them onto brain activity. Text captions, video frames, and audio are extracted and processed using pretrained models: XCLIP for text and video, and CLAP for audio, generating embeddings ($z_t$, $z_v$, $z_a$). These embeddings are then used in Ridge regression models to predict brain activity. The final step correlates the predicted brain activity with the actual brain data, visualized as brain maps. The frozen symbols indicate that encoder parameters are fixed during training, while Ridge regression models are trained for each modality.

## 2 Material and Methods

We analyzed a public available fMRI dataset acquired while ten subjects watched a video, originally released as part of the Algonauts 2021 challenge [6, 16]. In the main experiment, participants viewed 1,000 training videos and 102 testing videos multiple times, all presented without audio. MRI data were collected using a 3T Siemens Trio scanner, and preprocessing included standard fMRI procedures such as slice time correction and normalization. The data were part of the Algonauts 2021 challenge, focusing on 1,000 fMRI-video pairs. For more detailed information about the data collection and preprocessing, please refer to the appendix and the original article [6]. Our primary objective was to identify brain regions responsive to identify potentially distinct brain regions responsive to each specific modality. We extracted captions from videos using a video captioning model [25], along with video frames and audio. These stimuli representations (semantic, visual, and audio) were processed using pretrained computational models to obtain modality-specific embeddings. Transformer-based models, such as XCLIP [17, 20], were used to extract video and text embeddings, while CLAP [9] was employed to obtain audio embeddings. We then modeled the mapping between brain activity and embeddings using Ridge regression. For each modality, the encoding models took as input embeddings with a dimensionality of 512 and projected them onto estimated brain activity, which had a dimensionality corresponding to the number of voxels (ranging from 10,836 to 21,573, depending on the subject). All models were subject-specific and trained using nested 5-fold cross-validation to prevent any potential circularity with the decoding models. The inner loop was used for hyperparameter optimization, while the outer loop predicted held-out data from the training set. Once the full training set was predicted, Pearson correlation was computed along the samples dimension, resulting in a voxel-wise map of correlations between predicted and actual brain activity for each modality. A threshold of $0.15$, determined empirically, was then applied to create a ROI for each modality, which was used in subsequent analyses.We found "activations" in visual, auditory, language, and multimodal integration brain areas corresponds to video, audio, and text inputs, indicating distributed processing across both unimodal and multimodal regions. To account for variability in brain structure and function across individuals, we used a modality-specific functional alignment strategy within a 5-fold nested cross-validation framework. This approach aligns brain activity patterns across subjects for each modality (semantic, visual, and audio), maximizing the similarity of functional responses while retaining modality-specific information. Following recent literature [11, 7, 2], we employed ridge regression as a regularization technique to improve robustness and generalizability by addressing multicollinearity in the high-dimensional fMRI data. This alignment method aimed to enhance the performance of our decoding models across multiple

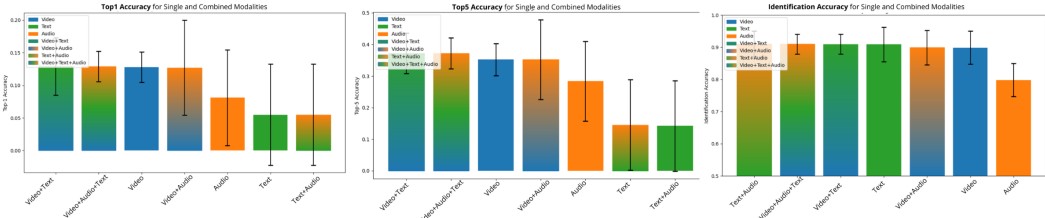

Figure 3: This figure presents three key performance metrics evaluating the effectiveness of decoding models across different single and combined modalities (Video, Text, Audio, Video+Text, Video+Audio, Text+Audio, and Video+Text+Audio). From left to right: top1, top5 and identification accuracy.

subjects. The above procedures resulted, for each modality, in a set of 9,000 training video-fMRI pairs (900 pairs per subject) and 1,000 testing pairs (100 pairs per subject). We then developed modality-specific decoding models by training cross-validated Ridge regression models within each identified ROI. These models were used to estimate the modality embeddings from brain activity (e.g., estimating text embeddings from regions responsive to text stimuli, and similarly for visual and audio modalities). For video retrieval, we employed a Euclidean search strategy, selecting the top-N closest test videos based on the L2 distance between the estimated and true embeddings for each modality. Additionally, we implemented a modality integration-based search by concatenating embeddings from different modalities, thereby allowing for a more comprehensive retrieval process that leverages the combined information from multiple sensory streams. As evaluation we report three metric useful to identity quality of retrieval and decoding. The first two are Top-1 and Top-5 accuracy, which simply count how many times the first retrieved videos is exactly the stimulus or when the the stimulus is correctly retrieved among the first 5 five retrieved videos. To complete the analysis and evaluate the quality of the decoded embeddings, we also report the identification accuracy, originally defined in [23]. This metric is a pairwise measure based on correlation, where a value of 0.5 indicates random predictions and a value of 1 signifies perfect predictions. The identification accuracy counts the number of times the estimated embeddings correlate more strongly with the true embedding than with other embeddings. This metric provides a robust evaluation of the quality of the estimated embeddings, which is crucial for more complex tasks such as generation.

# 3 Results

The results, summarized in Fig. 3 and Tables 2, 1, suggest that multimodal integration significantly enhances decoding and retrieval performance. On average, the Video+Text+Audio combination achieved the highest identification accuracy, around 0.94, consistently outperforming other combinations. Video+Text and Video+Audio combinations also performed well, with average accuracies of 0.92 and 0.91, respectively. All combinations performed above chance level, highlighting the effectiveness of multimodal integration. In terms of video retrieval, the Video+Text+Audio combination again provided the highest average Top-1 and Top-5 accuracies, confirming that combining multiple modalities provides best results for decoding and retrieval tasks.

# 4 Discussion and Conclusions

In this study, we explored the advantages of a multi-stream approach for decoding brain activity, focusing on integrating Video, Text, and Audio modalities. Combining multiple modalities enhances decoding and retrieval performance compared to using individual modalities. The Video+Text+Audio combination consistently outperformed other combinations, indicating that this multi-stream integration captures more comprehensive and complementary information.

To implement this approach, we leveraged state-of-the-art multimodal models like XCLIP and CLAP. Growing evidence suggests that multimodal models more accurately capture brain patterns than unimodal models, due to their ability to process richer and more informative embeddings [5, 18, 1]. These embeddings reflect the multimodal sensory integration mechanisms that occur in the brain, aligning with the brain's tendency to synthesize data from various senses to build

**Stimulus**                    **Retrieved**

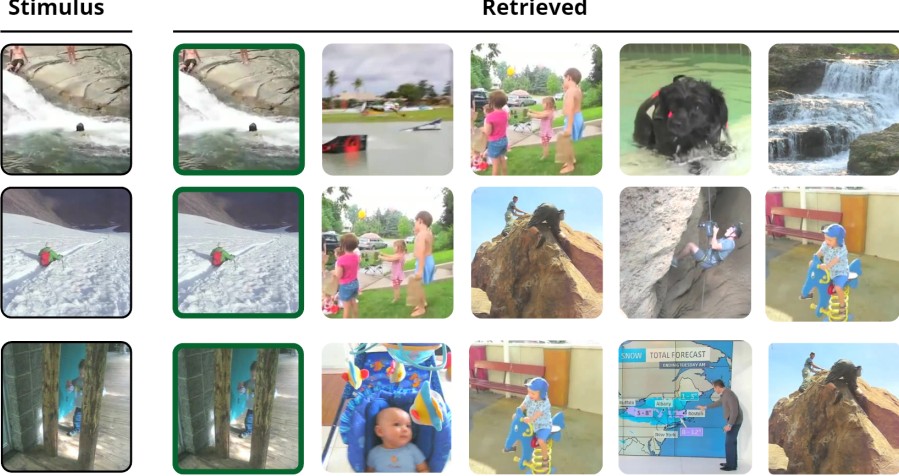

Figure 4: Some examples video stimuli (first frame) and retrieved pool of candidates from mixed modality Video+Audio+Text, subject01

cohesive representations. This integration is particularly advantageous in neuroimaging contexts, where understanding brain patterns relies on diverse information from multiple sensory domains.

Interestingly, the audio modality, despite the fact that videos were presented without sound, still performed above chance level in both identification and retrieval tasks. This suggests that the brain may integrate audio-visual information even in the absence of one modality, aligning with findings from hearing and optical illusions where the brain uses available sensory data to construct a complete perceptual experience. This observation underlines the complex and interconnected nature of sensory processing in the brain, where information from one modality can influence the processing of another, even when it is not directly presented.

This study is not without limitations. The reliance on pre-trained models for generating embeddings may not fully capture the specific neural representations relevant to each participant, potentially introducing model biases. As these models are trained on vast, diverse datasets, their internal representations may inadvertently reflect biases or inaccuracies that could misinterpret individual brain activity. This issue emphasizes the need to consider the ethical implications of decoding models, particularly when dealing with sensitive or personal neural data. Implementing safeguards and ethical guidelines, including transparent and interpretable models, can mitigate the risks of misinterpretation or misuse of brain decoding technology.

Looking ahead, a promising direction for future work involves the development of generative models that can not only decode brain activity into embeddings but also reconstruct the original stimuli. One potential pathway could involve a two-step approach: first, a multimodal encoding model could be developed, as we have done in this study. Then, building on prior work [10], brain activity could be decoded into a textual representation of the videos, which could be used alongside a generative model in combination with a Bayesian approach to generate videos that match the brain activity measured. This advancement would enable a more direct and intuitive understanding of how the brain encodes and processes complex sensory information, providing deeper insights into the neural mechanisms of perception.

As we advance in the field of brain decoding, the creation of generative models that reconstruct stimuli from brain data could bring transformative applications across various fields, but also underscores the importance of ethical considerations. Ensuring neural privacy and responsible use of decoding models is paramount, especially as this technology continues to mature.

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

# A Appendix / supplemental material

## A.1 Data details

The study involved ten participants, each undergoing five scanning sessions. The first session was a localizer experiment where participants passively viewed short videos (unrelated to the main experiment) to define visual ROIs. The remaining four sessions constituted the main experiment, where participants focused on a fixation cross while viewing 3-second training and testing videos without audio. Videos were presented in 13 runs per session, each lasting about 7 minutes. By the end, participants had viewed 1,000 training videos three times and another set of 102 videos ten times. Using data from the localizer experiment, nine non-overlapping ROIs were defined for each participant, covering regions from early visual cortex (V1, V2, V3, V4) to higher-level areas responding to objects and categories (EBA, FFA, STS, LOC, PPA). MRI data were collected on a 3T Siemens Trio scanner with consistent acquisition parameters across all sessions (TR = 1750 ms, resolution = 2.5 mm³, 54 slices, multi-band factor = 2). Preprocessing was done using fMRIprep, including slice time correction, realignment, co-registration, and normalization to MNI space. Data were interpolated from TR = 1750 ms to 1000 ms using the pchip method. FIR basis functions modeled the BOLD signal for each voxel, extracting beta values from 5 to 9 seconds post-video onset. These were averaged across time and used in subsequent analyses. Our study focused on data from the Algonauts 2021 challenge, specifically 1,000 fMRI-video pairs, with the first 900 used for training and the last 100 for testing.

## A.2 Subject performances detail

| Subject | Video Identification Accuracy | Text Identification Accuracy | Audio Identification Accuracy | Video Top-1 | Video Top-5 | Text Top-1 | Text Top-5 | Audio Top-1 | Audio Top-5 |
|---------|-------|-------|-------|-------|-------|-------|-------|-------|-------|
| sub01 | 0.952 | 0.953 | 0.881 | 0.300 | 0.570 | 0.110 | 0.260 | 0.190 | 0.450 |
| sub02 | 0.934 | 0.899 | 0.805 | 0.160 | 0.440 | 0.050 | 0.160 | 0.040 | 0.240 |
| sub03 | 0.949 | 0.922 | 0.805 | 0.160 | 0.460 | 0.070 | 0.130 | 0.070 | 0.280 |
| sub04 | 0.916 | 0.941 | 0.751 | 0.100 | 0.340 | 0.050 | 0.120 | 0.070 | 0.250 |
| sub05 | 0.885 | 0.891 | 0.785 | 0.080 | 0.270 | 0.050 | 0.130 | 0.050 | 0.260 |
| sub06 | 0.915 | 0.893 | 0.835 | 0.150 | 0.370 | 0.030 | 0.130 | 0.100 | 0.300 |
| sub07 | 0.822 | 0.870 | 0.810 | 0.040 | 0.170 | 0.030 | 0.070 | 0.100 | 0.300 |
| sub08 | 0.796 | 0.869 | 0.745 | 0.070 | 0.200 | 0.060 | 0.110 | 0.070 | 0.240 |
| sub09 | 0.940 | 0.952 | 0.781 | 0.130 | 0.430 | 0.060 | 0.160 | 0.080 | 0.290 |
| sub10 | 0.878 | 0.900 | 0.780 | 0.090 | 0.270 | 0.040 | 0.150 | 0.040 | 0.230 |

Table 1: Single Modality Results

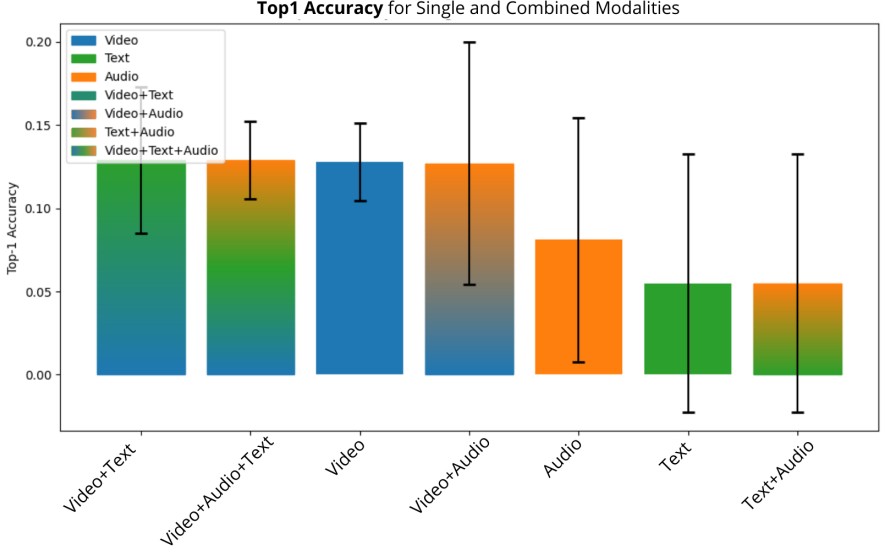

Figure 5: Identification accuracy, zoom of Right panel of fig 3

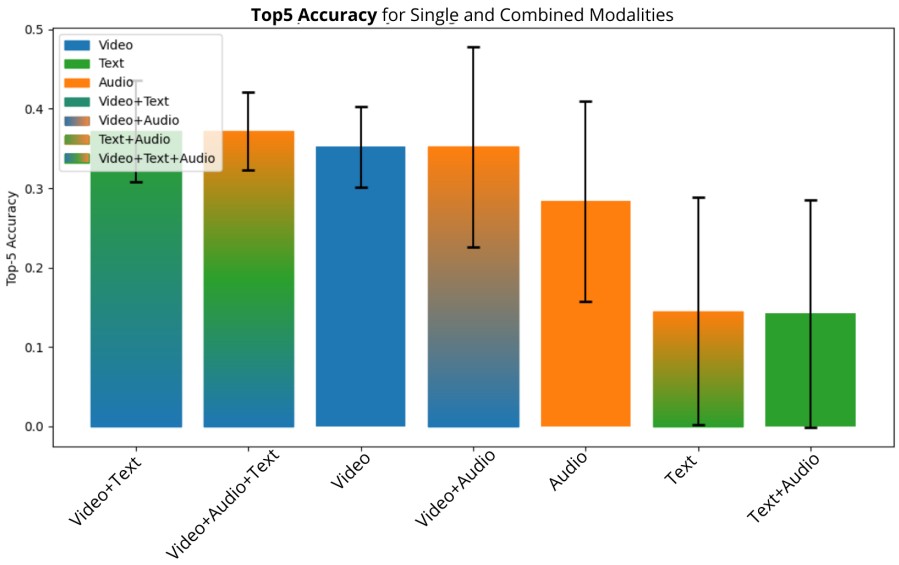

Figure 6: Top1 Accuracy, zoom of Left panel of fig 3

| Subject | Video + Text Id. Acc | Video +Audio Id. Acc | Text + Audio Id. Acc | Video +Text +Audio Id Acc | Video + Text Top-1 | Video +Text Top-5 | Video + Audio Top-1 | Video +Audio Top-5 | Text + Audio Top-1 | Text + Audio Top-5 | Video + Text + Audio Top-1 | Video +Text + Audio Top-5 |
|---|---|---|---|---|---|---|---|---|---|---|---|---|
| sub01 | 0.962 | 0.951 | 0.953 | 0.962 | 0.320 | 0.600 | 0.300 | 0.570 | 0.110 | 0.270 | 0.320 | 0.600 |
| sub02 | 0.943 | 0.934 | 0.901 | 0.943 | 0.100 | 0.470 | 0.160 | 0.440 | 0.050 | 0.160 | 0.100 | 0.470 |
| sub03 | 0.956 | 0.950 | 0.924 | 0.956 | 0.170 | 0.520 | 0.160 | 0.460 | 0.070 | 0.140 | 0.170 | 0.520 |
| sub04 | 0.931 | 0.916 | 0.942 | 0931 | 0.100 | 0.370 | 0.090 | 0.340 | 0.050 | 0.130 | 0.100 | 0.370 |
| sub05 | 0.895 | 0.885 | 0.893 | 0.895 | 0.110 | 0.300 | 0.080 | 0.270 | 0.050 | 0.130 | 0.110 | 0.300 |
| sub06 | 0.923 | 0.916 | 0.896 | 0.923 | 0.130 | 0.370 | 0.150 | 0.370 | 0.030 | 0.140 | 0.130 | 0.370 |
| sub07 | 0.834 | 0.823 | 0.873 | 0.834 | 0.040 | 0.140 | 0.040 | 0.170 | 0.030 | 0.070 | 0.040 | 0.140 |
| sub08 | 0.813 | 0.796 | 0.870 | 0.813 | 0.070 | 0.210 | 0.070 | 0.200 | 0.060 | 0.110 | 0.070 | 0.210 |
| sub09 | 0.950 | 0.940 | 0.953 | 0.950 | 0.160 | 0.460 | 0.130 | 0.430 | 0.060 | 0.150 | 0.160 | 0.460 |
| sub10 | 0.889 | 0.879 | 0.901 | 0.890 | 0.090 | 0.280 | 0.090 | 0.270 | 0.040 | 0.150 | 0.090 | 0.280 |

Table 2: Mixed Modality Results

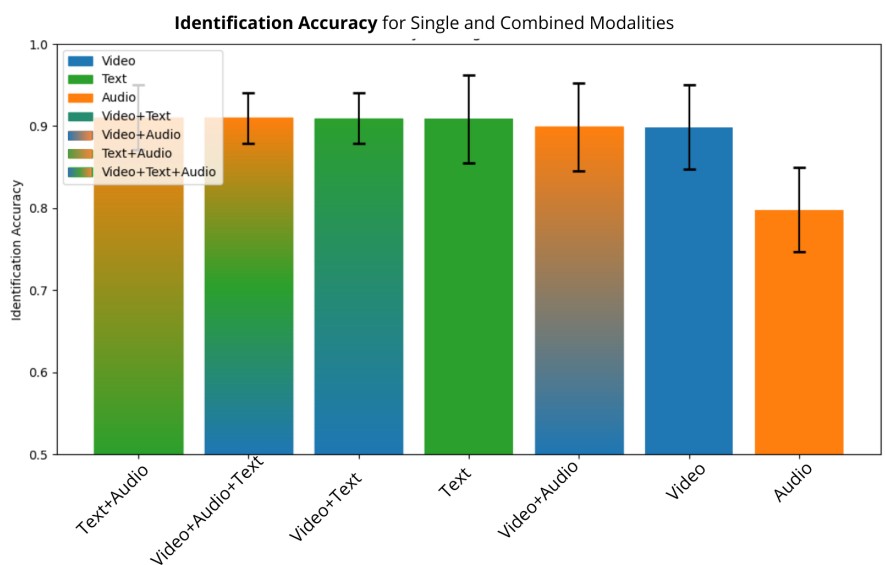

Figure 7: Top5 Accuracy, zoom of Center panel of fig 3

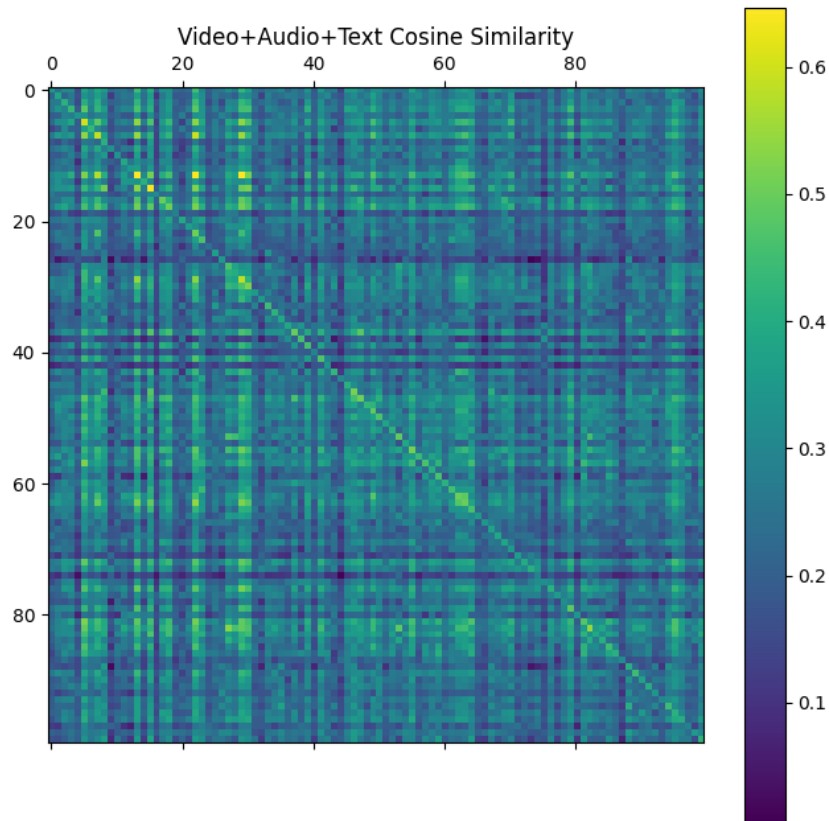

Figure 8: Average Cosine similarity matrix between true and predicted test embeddings.

**Stimulus**          **Retrieved**

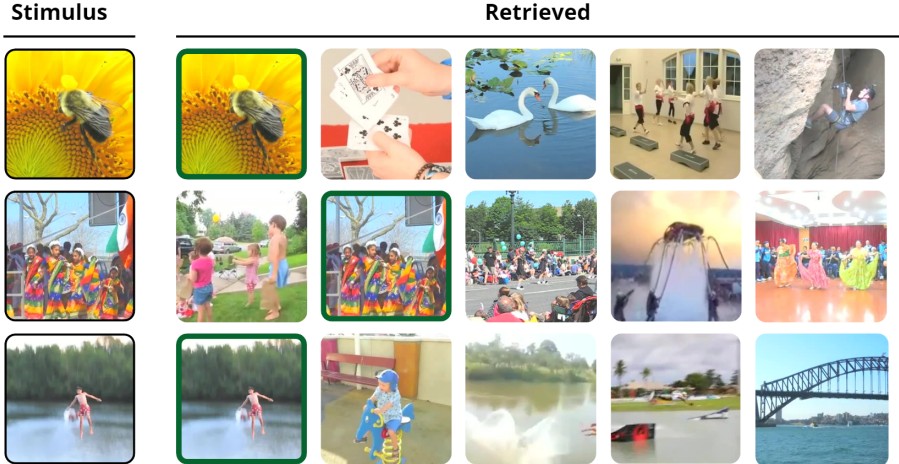

Figure 9: Some examples video (first frame) stimuli and retrieved pool of candidates from mixed modality Video+Audio+Text, subject01

**Stimulus**          **Retrieved**

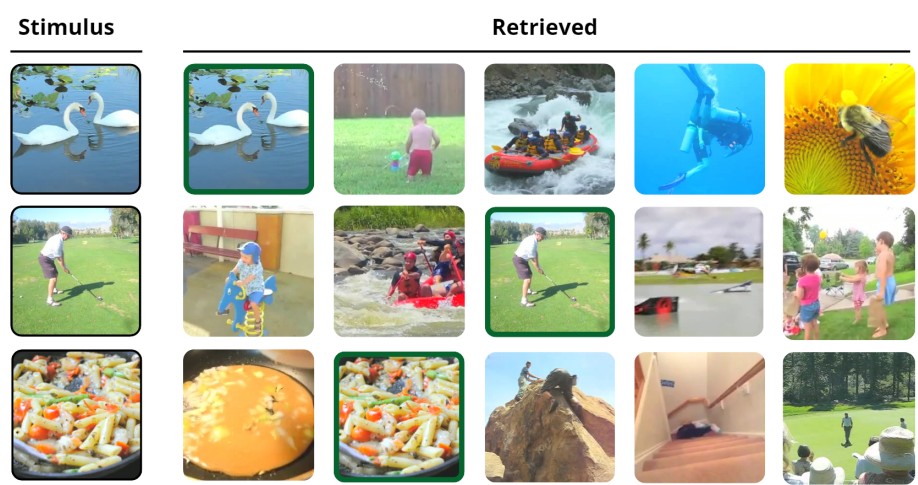

Figure 10: Some examples video stimuli (first frame) and retrieved pool of candidates from mixed modality Video+Audio+Text, subject01

