# OpenReview forum: "Video decoding from human fMRI data with a multi-stream sensory approach"
_NeurIPS.cc/2024/Workshop/UniReps — UniReps_

### Official Review · Reviewer_M86Q · 2024-09-28
**While this paper presents a solid multi-stream sensory approach for decoding video stimuli from fMRI data by combining visual, textual, and audio modalities, the conclusion that fusing all three improves performance is somewhat predictable. It is intuitive that more information from additional modalities would improve decoding accuracy.**

**Rating:** 6
**Confidence:** 5

**Review:**

### Quality
The paper is of relatively high quality in terms of methodology and implementation. It presents a robust pipeline for decoding video stimuli from fMRI data using a multi-modal approach incorporating video, text, and audio embeddings.

### Clarity
The paper is clear and well-structured. The authors provide a thorough explanation of their decoding pipeline, including visual aids that enhance understanding. The step-by-step breakdown of the encoding and retrieval processes is well-presented.

### Originality
While the multi-modal integration of video, text, and audio in fMRI decoding is interesting, it is not a groundbreaking innovation. The conclusion that combining more sensory streams improves decoding performance is intuitive.

### Pros:
I liked two things about this paper:

1. They proposed a novel approach by integrating multiple modalities (video, text, audio) for fMRI-based video decoding. The inclusion of these different sensory streams helps improve decoding accuracy, which is a step forward from single-modality approaches.
2. The cross-subject functional alignment strategy enhances model generalization, which is crucial for building robust brain decoding models across individuals.

### Cons:
I have a few significant concerns as well:

1. They rely on pre-trained models (XCLIP, CLAP) for generating embeddings, which may not fully capture the specific neural representations relevant to individual participants. Instead of using pre-trained models as-is, they can be fine-tuned with fMRI data specific to each individual.
2. Although the paper mentions the potential of generative models for future work, it doesn't explore them in this study. Given the current trend toward using diffusion models or GANs in fMRI decoding, this could be seen as a missing aspect of the current work.
3. Although the results demonstrate that fusing three modalities improves performance, this finding is somewhat predictable and not novel.

---

### Official Review · Reviewer_SqYw · 2024-09-30
**Multimodal Decoding of fMRI Data: A Peer Review of Video Decoding from Human fMRI Using a Multi-stream Sensory Approach**

**Rating:** 9
**Confidence:** 4

**Review:**

## Strengths:

### 1. Innovative Multimodal Approach
The paper presents a novel approach to decoding brain activity by integrating multiple sensory streams—visual, textual, and audio modalities—using fMRI data. This multimodal framework allows for a more comprehensive understanding of how different stimuli are processed by the brain, significantly improving the performance of decoding models. The combination of these modalities (Video+Text+Audio) achieves the highest identification and retrieval accuracy, which is a major advancement in this field.

### 2. Subject-Specific Encoding Models
The study uses subject-specific models to predict brain activity based on modality-specific embeddings, enhancing accuracy and generalizability. The functional alignment technique across subjects further strengthens the robustness of these models, allowing for better cross-subject comparisons and consistency.

### 3. Thorough Evaluation Metrics
The use of Top-1 and Top-5 accuracy, alongside identification accuracy metrics, provides a clear and comprehensive evaluation of the decoding framework. The high performance of the multimodal integration models (Video+Text+Audio) across all metrics demonstrates the effectiveness of the proposed approach.

### 4. Data Collection and Processing
The paper utilizes a well-structured dataset from the Algonauts 2021 challenge, including detailed fMRI data from multiple subjects. Preprocessing steps such as slice time correction, normalization, and functional alignment were well executed, which adds to the credibility of the results.

## Weaknesses:

### 1. Over-reliance on Pretrained Models
The approach heavily depends on pretrained models (XCLIP for video and text embeddings, CLAP for audio), which might not fully capture the neural representations specific to each participant. While this provides a good starting point, it limits the study's ability to dive deeper into individual differences in brain activity and how specific features of stimuli are processed.

### 2. Audio Modality Ambiguity
The paper reports that even though the videos were presented without sound, the audio modality still performed above chance level. While this observation is intriguing, the explanation for this phenomenon remains speculative and lacks sufficient theoretical discussion. A more in-depth exploration of why the audio modality contributes to decoding without sound stimuli would strengthen this part of the paper.

### 3. Generative Model Limitations
The study hints at the potential future application of generative models to reconstruct stimuli from brain activity. However, no significant steps are taken towards implementing or testing such models in the current work. While the focus is on retrieval and identification, the paper could have explored initial steps towards reconstruction, given that this is a key interest in brain decoding research.

### 4. Limited Dataset
Although the dataset of 1,000 videos is substantial, a broader range of stimuli, particularly in terms of diversity in content (beyond short clips), would make the findings more generalizable. Additionally, while the authors use functional alignment across subjects, the dataset still comes from a relatively small number of subjects (10), which may limit the conclusions regarding cross-subject generalizability.

## Suggestions for Improvement:

- **Address the Pretrained Model Limitation**: A deeper exploration of the limitations of using pretrained models for extracting embeddings, as well as potential alternatives or custom approaches, would enhance the study’s depth. Integrating models trained specifically on brain data could yield better alignment with neural representations.

- **Explore the Audio Phenomenon Further**: The unexpected performance of the audio modality without sound input is fascinating. A more detailed investigation into this finding, potentially supported by additional experiments, could add an important dimension to the paper’s contributions.

- **Progress Toward Generative Models**: While retrieval and identification are well-handled, the paper would benefit from an initial exploration of generative models that can reconstruct the original stimuli from brain activity, as mentioned in the conclusion. Even preliminary experiments in this area would provide a roadmap for future work.

- **Expand Dataset and Modality Range**: Incorporating a more diverse set of video clips or additional sensory modalities could further strengthen the findings. This would also offer more insights into how different types of stimuli are processed in the brain, beyond the specific setup used in this study.

## Conclusion:
This paper provides a novel and well-executed contribution to the field of brain decoding by integrating multimodal sensory streams and leveraging subject-specific encoding models. The approach shows strong potential for improving decoding accuracy, particularly with the combination of video, text, and audio modalities. However, there are some limitations, such as the reliance on pretrained models and the unexplained performance of the audio modality without sound. Further work could explore generative models and larger, more diverse datasets to expand on these promising results.

---

The paper is a solid contribution, with a clear path forward for improvements and future research directions.

---

### Official Review · Reviewer_snK7 · 2024-10-02
**The paper tackles an interesting problem by proposing a multi-modal approach for decoding video stimuli from fMRI data, integrating visual, text, and audio modalities. The framework shows potential, but the experimental setup and evaluation need more precision and depth to validate the robustness of the method. Improvements are needed in figure clarity, label accuracy, metric justification, and experimental design. Additionally, the lack of comparison with existing methods limits the assessment of the approach’s novelty. Addressing these concerns would strengthen the paper’s contribution to the field of brain decoding.**

**Rating:** 4
**Confidence:** 4

**Review:**

The paper addresses an interesting problem by proposing a multi-modal approach to decode video stimuli from fMRI data, integrating visual, text, and audio modalities. The authors attempt to demonstrate the effectiveness of their framework, using Ridge regression and functional alignment to improve identification and retrieval accuracy. While the results are promising, the experimental setup and evaluation require more precision and depth to fully confirm the robustness of the approach. A clearer experimental design and more thorough evaluation metrics would strengthen the claims and provide a more concrete validation of the method. There are some suggestions for possible improvements:

1.	The labels in Figure 1 are too small and not clearly visible, and some x-axis labels are cropped, making the figure difficult to interpret.

2.	The labels in Figures 4, 5, and 6 are incorrect and need to be revised.

3.	The visualization and coloring in Figure 3 are confusing, and some labels are not fully shown. This figure needs better clarity in terms of coloring and label placement.

4.	Figure 2 lacks a colorbar, which is crucial for understanding the range of correlations, and only cortical voxels are shown without discussion of subcortical voxel activities.

5.	The identification accuracy metric may not fully capture model performance. Although the embeddings may correlate more strongly with the true embeddings than others, the overall correlation could still be low, as seen in the top-1 and top-5 accuracies in Figure 3.

6.	The terms Video+Audio+Text are confusing because video inherently contains visual and audio elements. More appropriate terms would be Visual, Text (Semantic), and Audio modalities to clearly distinguish the different streams.

7.	The encoding step requires learning a large number of parameters, yet the paper does not discuss the number of samples used for learning these parameters, which is an important consideration for model robustness.

8.	There is a misplaced ")" in line 8 of the abstract, and a typo exists in line 42 that should be corrected.

9.	The paper does not present significance measures for the accuracy distributions, making it difficult to determine whether there are statistically significant differences in performance when recovering videos using different combinations of modalities (e.g., comparison between the first four categories from the left in Figure 3).

10.	There is no comparison with existing multi-modal brain decoding methods or simpler baselines, making it difficult to assess the true novelty and improvement of the proposed approach. Including comparisons with previous methods would strengthen the claim of novelty and demonstrate the advantage of the current approach.

---

### Official Review · Reviewer_xumu · 2024-10-06
**Innovative Multi-Modal Approach to Video Decoding from fMRI Data: A Comprehensive Evaluation**

**Rating:** 8
**Confidence:** 5

**Review:**

The manuscript presents a high-quality and thorough methodology for decoding video stimuli from human fMRI data using a novel multi-stream sensory approach. The authors effectively utilize a substantial dataset consisting of 1,000 short video clips paired with corresponding fMRI data collected from ten subjects. Their approach incorporates advanced computational techniques, including Ridge regression, which serves as a regularization method to mitigate multicollinearity issues that are often prevalent in high-dimensional neuroimaging data. The study notably employs functional alignment methods across subjects, allowing for the normalization of brain activity patterns to maximize the robustness of the derived models. By identifying regions of interest (ROI) specific to each modality—video, audio, and text—the authors enhance the predictive accuracy of their subject-specific encoding models. The results convincingly indicate that the integration of multiple sensory modalities significantly improves decoding performance, particularly with the Video+Text+Audio combination achieving a top identification accuracy of approximately 0.94, outpacing other combinations such as Video+Text and Video+Audio.

The clarity of the writing is generally strong, with a systematic structure that guides readers from a foundational introduction to complex methodology, results, and conclusions. However, more precise elaboration on the technical aspects of the methods would improve the manuscript's accessibility. For instance, the authors employ state-of-the-art transformer architectures, such as XCLIP for generating embeddings from video and text, and CLAP for audio embeddings. A deeper explanation of how these models are fine-tuned to capture modality-specific features would be beneficial. Additionally, the nested 5-fold cross-validation technique is a robust choice for hyperparameter optimization and validation, yet a detailed discussion on the process of aligning functional responses across subjects—possibly through techniques like non-linear transformations or the use of technique-specific distance metrics—would enhance understanding.

Overall, while the manuscript is well articulated, addressing these specific technical aspects would ensure that it is engaging to a wider audience, including those less familiar with advanced computational techniques.In terms of originality, this study makes a significant contribution to the field of brain decoding by pioneering the multi-sensory integration approach to enhance decoding performance from fMRI data. The hypothesis positing that video processing can be decomposed into distinct streams—visual, semantic, and auditory—is supported by the findings that demonstrate nuanced performance across different modality combinations. Notably, the manuscript reveals how the auditory modality, despite being absent during video presentation, still provides above-chance levels of identification and retrieval accuracy, suggesting that the brain engages in predictive processing even when certain sensory information is not directly available. Furthermore, while the ethical implications of neural privacy are briefly addressed, a thorough exploration of how biases inherent in pre-trained models, such as those used for generating video, text, and audio embeddings, might skew individual neural representation would strengthen the discussion. It is crucial to consider how algorithmic biases can lead to misinterpretations, especially as decoding technologies continue to evolve, necessitating a framework for ethical use and transparency in the application of these advanced techniques.

---

### Decision · Program_Chairs · 2024-10-10

**Decision:**

Accept

**Comment:**

In light of the positive reviewers' feedback and relevancy of the submission, we are pleased to accept this paper for presentation at UniReps 2024. We kindly ask the authors to incorporate the reviewers' suggestions and feedback in the final camera-ready version of the manuscript.